# Detection of Black Spot of Rose Based on Hyperspectral Imaging and Convolutional Neural Network

**Jingjing Ma, Lei Pang, Lei Yan \* and Jiang Xiao**

School of Technology, Beijing Forestry University, No. 35 Tsinghua East Road, Beijing 100083, China; 17801034752@163.com (J.M.); panglei@bjfu.edu.cn (L.P.); xiaojiang@bjfu.edu.cn (J.X.)

\* Correspondence: mark_yanlei@bjfu.edu.cn; Tel.: +86-010-62336913

**Abstract:** Black spot is one of the seriously damaging plant diseases in China, especially in rose production. Hyperspectral technology reflects both external features and internal structure information of measured samples, which can be used to identify the disease. In this research, both the spectral and image features of two infected roses with black spot were used to train a convolutional neural network (CNN) model. Multiple scattering correction (MSC) and standard normal variable (SNV) methods were applied to preprocess the spectral data. Cropping, median filtering and binarization were pretreatments used on the hyperspectral images. Three CNN models based on Alexnet, VGG16 and neural discriminative dimensionality reduction (NDDR) were evaluated by analyzing the classification accuracy and loss function. The results show that the CNN model based on the fusion of features has higher accuracy. The highest accuracies of detection of blackspot in different roses are 12–26 (100%) and 13–54 (99.95%), applying the NDDR-CNN model. Therefore, this research indicates that the spectral analysis based on CNN can detect black spot of roses, which provides a reference for the detection of other plant diseases, and has favorable research significance as well as prospect for development.

**Keywords:** hyperspectral imaging; black spot; infected samples; non-destructive detection; preprocessing; CNN model

---

## 1. Introduction

Roses have great commercial value as ornamental plants, food and exports as cut flowers in China [1]. Black spot is one of the most severe and devastating disease of roses [2]. It is caused by black spot fungus. Symptoms of infection mainly include round spots with a black and feathery edge on the front side of the leaves [3]. At present, the method to manage the disease is to use fungicides or pesticides. The primitive means of detection depends on the visual ratings of gardeners after spot spreading which is inefficient and inaccurate [4,5]. If the disease can be accurately identified in the early stage, it can be controlled in a timely manner and improve economic benefits. Therefore, it is urgent to comprehensively improve the detection ability of black spot disease.

Some biological methods are also used to detect plant diseases. Ju et al., developed an assay consisting of recombinase polymerase amplification combined with lateral-flow dipstick technology (RPA-LFD) for the rapid and sensitive detection of V. dahliae [6]. Shi et al., applied loop-mediated isothermal amplification (LAMP) to detect P. carotovorum in celery with soft rot using a primer set designed from the pmrA conserved sequence of P. carotovorum [7]. PCR analysis was used to identify and detect powdery and downy mildew on cucumber as well as Pectobacterium brasiliense on potato [8,9]. These research methods have the advantages of rapid speed and sensitivity, but also have shortcomings, such as being expensive, destructive to the leaves and requiring a cumbersome operation process and professional biochemical knowledge. Hyperspectral technology is an advanced technology which combines the traditional spectral technology

and two-dimensional imaging technology organically. It can reflect both external feature information and internal structure information of the samples. Compared with manual and biological detection, it has the characteristics of being fast, non-destructive, non-polluting and easy to operate. It has been gradually applied in related fields and it proposes a potential solution to solve many problems faced by human visual-level detection of plant pathology in the field. Roscher et al., combined the information of hyperspectral features with 3D geometry features to detect leaf spot in Cercospora [10]. Ban et al. studied the SPAD value of apple leaf infected with apple mosaic virus using hyperspectral transmission measurement technology [11]. Mahlein et al. summarized the hyperspectral imaging technology used to evaluate the relationship between plants and pathogens [12]. Mehrubeoglu et al. focused on the hyperspectral images of grape leaves to identify the red blotch disease and different infected stages of the disease [13]. Laurel wilt disease of avocado, early Ganoderma boninense disease of oil palm trees, fusarium head blight in wheat kernels and black Sigatoka in banana leaves in the early stage were detected automatically based on hyperspectral imaging [14–17]. Wahabzada et al. presented several data mining techniques applied to discover the spectral characteristics of some specific diseases [18]. Hariharan et al., developed a novel method to analyze hyperspectral data using finite difference approximation (FDA) and bivariate correlation (BC) to distinguish laurel wilt infected avocado from healthy trees with an overall accuracy of 100% [19]. Gradually, hyperspectral imaging and machine learning were used in many studies to detect the symptoms before disease. Zhang et al. chose the classification and regression tree (CRT) algorithm to establish the prediction spectral model of wheat powdery mildew, considering the effects of wheatear and leaf shadow [20]. The K-nearest neighbor (KNN) method was used to establish the discriminant models to classify healthy and gray mold infected tomato leaves and muskmelon Cercospora leaf spot with hyperspectral imaging techniques [21,22]. Chen et al. established and selected the most appropriate leaf-level reflectance-based vegetation indices for bacterial wilt detection in peanuts. ANOVA, multilayer perception, and the reduced sampling method were used to analyze the spectral data [23]. Garhwal and Park et al. developed a partial least squares discriminant analysis (PLS-DA) model to predict Zebra Chip in potatoes, Marssonina blotch in apple leaves, oak wilt and yellow rust on wheat leaves, respectively. The spectral signatures were extracted by segmentation and morphological operations [24–27]. Zhang and Pan et al., obtained hyperspectral images of rice leaves and pear fruit infection. A support vector machine (SVM) model was constructed to identify different infection severities based on the transformed data [28,29].

Among the existing methods for detecting plant diseases, the convolutional neural network (CNN) rarely appears. These methods achieve early detection of plant diseases, but there is still space for improvement in accuracy. In addition, the classic machine learning algorithm usually requires complex feature engineering, while CNN does not. It only needs to input the data directly to the network, which usually achieves good performance.

This study used hyperspectral imaging to detect black spot in two varieties of roses. Spectral and images were extracted and different CNN structures were adopted. The accuracy and efficiency showed that CNN had great potential in detecting plant diseases.

## 2. Materials and Methods

### 2.1. Hyperspectral Imaging System

The hyperspectral imaging system is shown in Figure 1. It contains an SOC710VP HS line-scanning imaging spectrograph (AZUP Scientific Ltd., Beijing, China) which is fixed to a bracket, a C type infrared correction lens, an object stage, a notebook computer with HyperScanner_2.0.127 software (SURFACE OPTICS CORPORATION, San Diego, CA, USA) for collecting the image, and two 150 W tungsten halogen lamps that can provide stable and multi band light. The imager extracts 128 wavebands with a spectral range of 370–1042 nm, a spectral resolution of 4.7 nm and a spatial resolution of 696 × 520 pixels. The exposure time of the imaging system was set to 3 ms. The location and elevation of lights were adjusted according to the imaging result of samples at a 45° angle, 65 cm away from the stage.

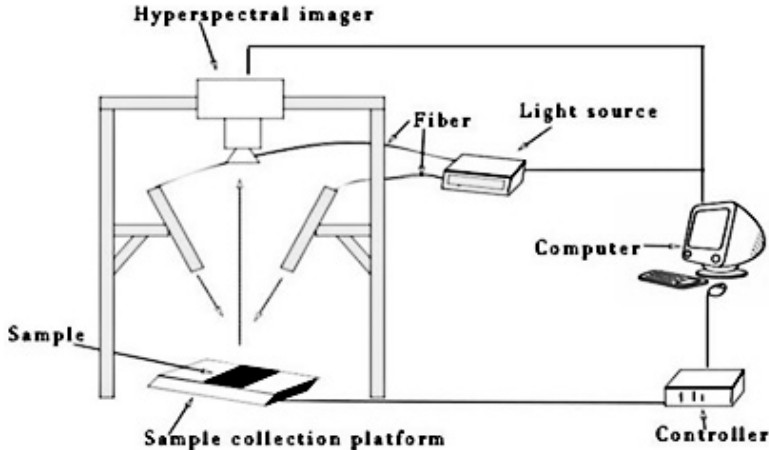

**Figure 1.** Hyperspectral imaging system.

*2.2. Plant Samples*

This study selected two kinds of roses consisting of 12–26 (susceptible) and 13–54 (resistant) which were both hybridization samples from the rose spot breeding laboratory in the School of Landscape Architecture. 90 plants of each kind were planted in a greenhouse with temperature (20 °C), humidity (60% relative), light (12 h) and an identical environment. Gathering five real leaves from the same position in the stem 20 cm from the ground where petiole length ranges from 2 to 4cm. Per kind, roses containing 450 leaves and 900 healthy leaves in total were set as the control group. The mycelium was scraped from the naturally occurring plants and prepared as a spore suspension with the potency of 0.5\1\1.5 mol/L under the microscope. Each kind of rose was divided in three groups and each group was inoculated with a concentration of suspension. Four bacterial drops were inoculated symmetrically on four positions on one leaf. It should be noted that the inoculation was carried out after the collection of the spectral data of all healthy leaves. Then all the inoculated leaves were put into the incubator to create conditions of temperature (25 °C), humidity (80% relative) and lucifugal for the growth of the black spot fungus. These were taken out every 24 h to collect spectral data until they were cultivated, at 7 days. After three weeks of continuous culture, the incidence of black spot fungus on each leaf was recorded and we divided the leaves into two levels of health and infection according to whether there were spots on the surface, as the label for later training. The samples after infection are shown in Figure 2. They were then put in a petri dish lined with filter paper and culture fluid. There was no obvious abnormality observed on the surface at the beginning of the inoculation.

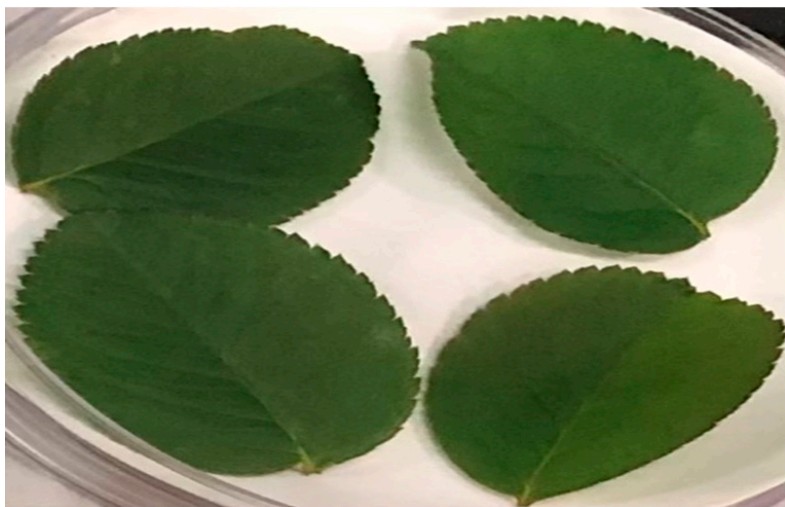

**Figure 2.** Rose leaves infected with black spot fungus.

## 2.3. Hyperspectral Imaging Acquisition and Calibration

Ambrose et al. put forward that the acquired raw hyperspectral images may be unavailable under the factors such as systematic noise and environmental influence [30]. Therefore, dark and white calibration for the images were needed. The calibration was achieved according to Equation (1)

$$I_C = \frac{I_r - I_d}{I_w - I_d} \tag{1}$$

where $I_c$ is the calibrated reflectance image; $I_r$ is the raw hyperspectral image; $I_d$ is the hyperspectral image of dark reference, which has almost 0% reflectance; and $I_w$ is the white reference hyperspectral image, which has a reflectance of over 99%.

## 2.4. CNN Models for Detection

In this study, three CNN structures are used for model training.

a. AlexNet

AlexNet was designed by ImageNet competition champion Hinton and his student Alex Krizhevsky in 2012. Compared with traditional machine learning classification algorithms, the main new technology points used by AlexNet are (1) successfully used ReLU as the activation function of CNN, and verified that its effect surpassed Sigmoid in the deeper network, and successfully solved the gradient dispersion problem of Sigmoid in the deeper network, (2) dropout is used to randomly ignore some neurons during training to avoid overfitting the model. In AlexNet, the last few fully connected layers use dropout, and (3) data enhancement. If there is no data enhancement, only the original data volume, CNN with many parameters will fall into overfitting. After using data enhancement, it can greatly reduce overfitting and improve the generalization ability.

The structure used in this study is as Figure 3. It contains 5 convolutional layers, 3 fully connected layers and 3 pooling layers. It is of most importance that 2 dropout layers are added to prevent overfitting. In this study, the input to this network is the spectral feature.

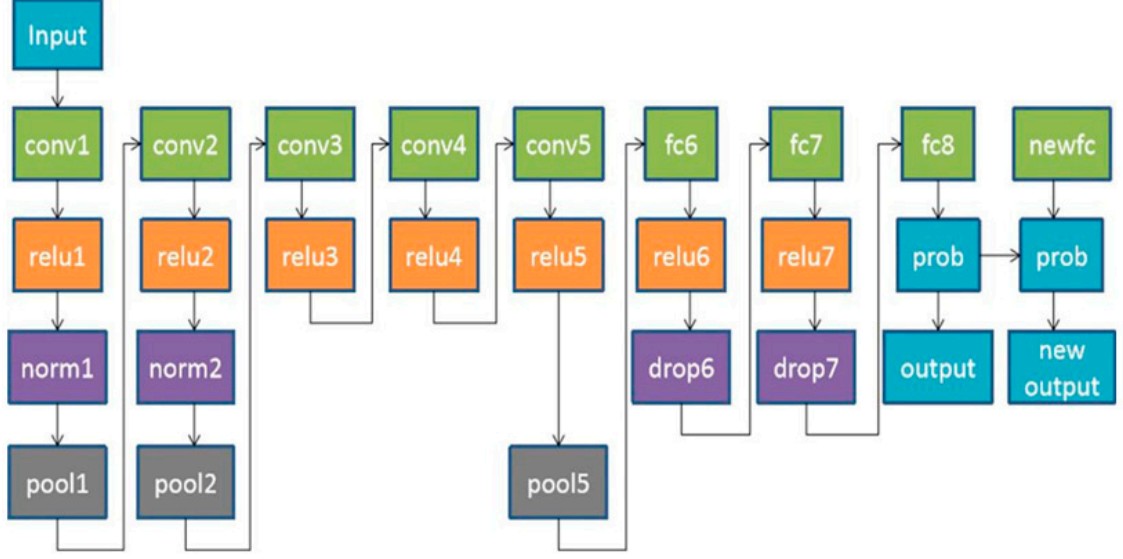

**Figure 3.** The architecture of AlexNet with dropout.

b. VGG16

VGG is a CNN model proposed by Simonyan and Zisserman in the document "Very Deep Convolutional Networks for Large Scale Image Recognition". The model participated in the 2014

ImageNet image classification and positioning challenge, ranking second in the classification task and first in the positioning task. The features of VGG are:

(1) Small convolution kernel. The author replaced all convolution kernels with $3 \times 3$ (rarely used $1 \times 1$).
(2) Small pooled core. Compared with AlexNet's $3 \times 3$ pooled cores, VGG are all $2 \times 2$ pooled cores.
(3) Fully connected to convolution. The network test phase replaces the three full connections in the training phase with three convolutions. The test reuses the parameters during training, so that the full convolutional network obtained by the test does not have the limit of full connection, so it can receive any width or height input.

According to the size of the convolution kernel and the number of convolution layers in VGG, it can be divided into 6 configurations (ConvNet configuration), A, A-LRN, B, C, D, and E; D and E are more commonly used and they are called VGG16 and VGG19.

In this study, VGG16 was selected as the training model; Figure 4 shows the structure. It contains 13 convolutional layers, 3 fully connected layers and 5 pooling layers. The input to this network structure is the hyperspectral image feature.

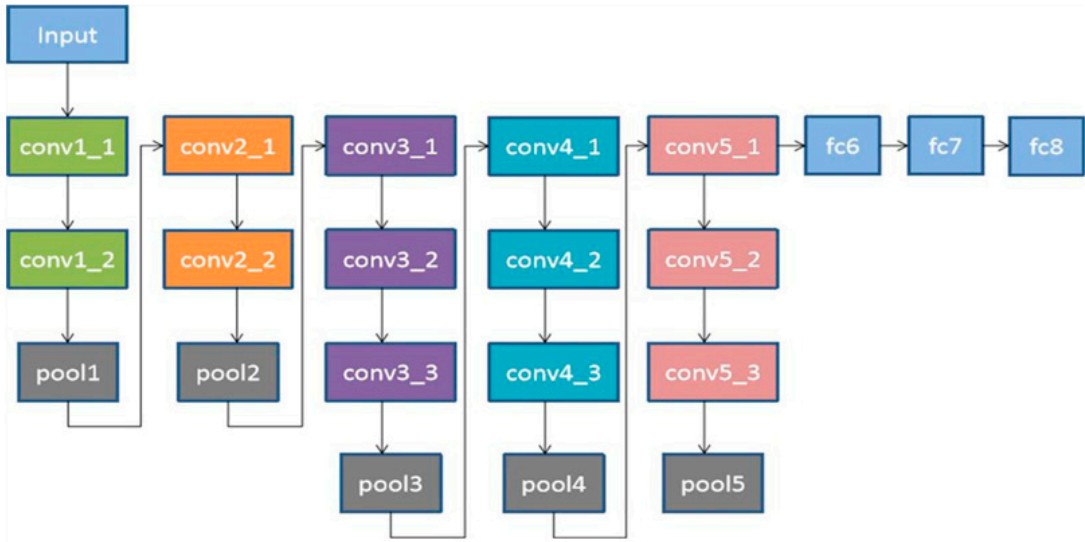

**Figure 4.** The basic structure of VGG16.

c. Neural discriminative dimensionality reduction (NDDR)-CNN

NDDR-CNN was proposed by Yuan Gao and others in 2019. It is a general-purpose multi-task CNN learning framework that can automatically integrate features of different layers of different tasks using the NDDR module, that is, no artificial hard design is required, which can achieve plug-and-play.

(1) NDDR layer. Used for multi-task feature fusion and feature dimensionality reduction. When the features of different layers of multiple tasks enter the NDDR layer, NDDR will first stitch all the incoming features in the last dimension, and then convolve the obtained features separately for each task. After completing the convolution, the obtained feature shapes are respectively input into the original network for convolution operation.
(2) Shortcuts. In order to prevent the gradient of the lower layer from disappearing, the Shortcuts module is used to directly pass the gradient from the last layer to the lower layer. Each mainline task will receive the feature from the NDDR layer multiple times. The Shortcuts layer of each task resplices multiple NDDR-features received by this task according to the last NDDR-feature and then stitches them together.

Experiments prove that the multitasking framework with NDDR has a certain degree of improvement compared to other frameworks. In addition, the task also processed many details, including the initialization of the weight of the NDDR layer and the selection of the learning rate, etc., which are all ways from which the network can learn.

The network structure of NDDR-CNN is shown in Figure 5. It uses the NDDR layer cascade to achieve multitasking. The input to this network structure in this study is a fusion of features, including spectral features and image features.

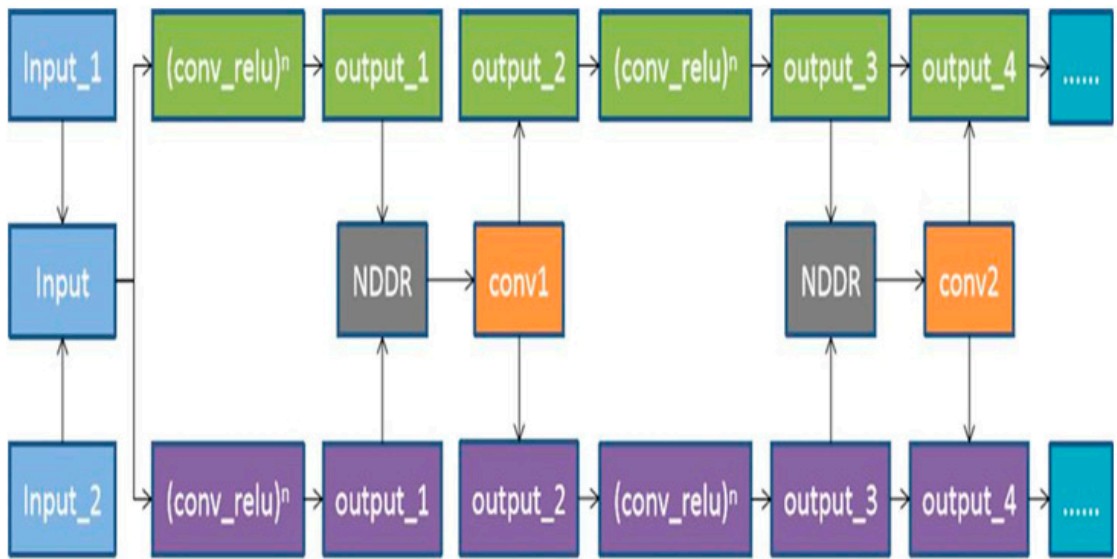

**Figure 5.** Neural discriminative dimensionality reduction (NDDR)-CNN architecture for black spot detection.

## 2.5. Data Processing

This study is based on Python 3.8 and processes the data in the environment of Pycharm + Tensorflow 2.1.0 + Keras 2.3.1. The data processing is as Figure 6.

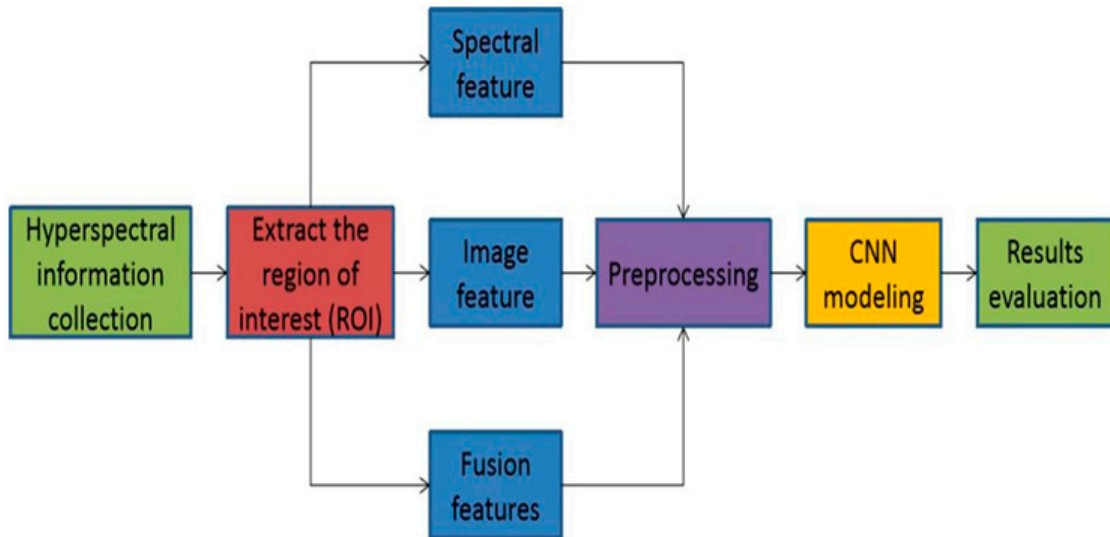

**Figure 6.** Schematic overview of the data procedures.

First of all, we obtained hyperspectral images including ordinary roses and those were inoculated with black spot. Secondly, image calibration was achieved by subtracting images of dark reference which were acquired at absolutely no light. The third step was to extract the region of interest (ROI)

from the hyperspectral images and extract the spectrum of ROIs, as is shown in Figure 7a. The average spectral of the sample points from each ROI was used to represent the spectrum of the ROI. The image data is also affected by the background and noise of the petri dish, so the image is preprocessed by cropping, median filtering denoising, and binarization. The processing flow and results are shown in Figure 7b. Finally, we chose 3 × 3 filtering to denoise the image, because this filter can detect more noise features and its denoising effect is better. For each waveband, 360 spectral samples obtained from the ROIs of each hyperspectral image were used as training sets for CNN modeling. After obtaining the CNN model, 90 new samples were used as the testing sets (Table 1). Modeling was then performed, based on spectral features, image features, and fusion features, respectively. Spectral features represent the reflectivity of the leaf to light at different wavelengths. Image features represent the grayscale pictures of the samples, containing some shape and texture features. When training the model, we used the spectrum, the image and a combination of the two as input. Finally various parameters of different CNN models were adjusted according to the training process, to obtain the optimal detection model.

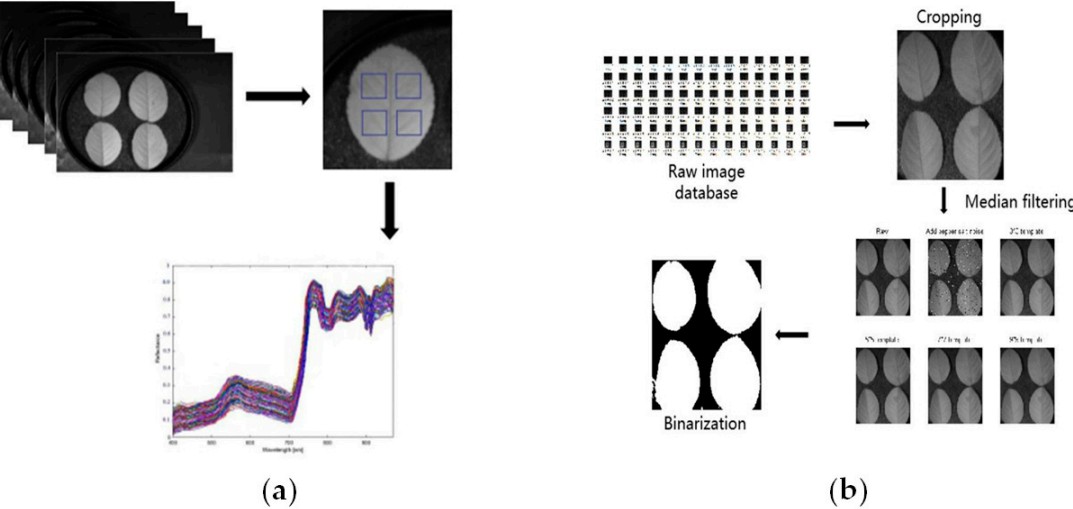

**Figure 7.** (**a**) Spectra extraction from ROIs; (**b**) noise removal from hyperspectral images.

**Table 1.** Information regarding healthy and infected samples for modeling.

| Variety | Treatments | Training Size | Testing Size |
|---|---|---|---|
| | Health | 360 | 90 |
| 12–26(Susceptible) | Infection | 360 | 90 |
| | Total | 720 | 180 |
| | Health | 360 | 90 |
| 13–54(Resistant) | Infection | 360 | 90 |
| | Total | 720 | 180 |

*2.6. Analysis*

As can be seen from Figure 7a, the spectral reflectance of different samples differs in value, but their changing trends are consistent, with the same absorption peaks and valleys. After extracting the ROI, we take the average spectrum of the sample as the research object. From Figure 8a, we see that the spectral reflectance of the infected sample is always lower than that of the healthy sample. At the same time, with the increase in the number of infection days, that is, the degree of infection continued to decrease, the reflectivity continued to decrease, showing such a trend in both roses. On the other hand, the spectral curve reflects the light absorption and reflection characteristics of rose leaves. At 580 nm, there is an obvious absorption peak, which is caused by the nitrogen response of the material in the leaf. The overall low reflectance within 800 nm is caused by the strong absorption of pigments (chlorophyll, anthocyanin, carotene, etc.) by the leaves. Within 800–1000 nm, the spectral reflectance increases

sharply, and then maintains a higher reflectance, which is caused by the multiple scattering of light by the leaf cells. The cell structure of the infected sample is destroyed, and the scattering ability is reduced, so the reflectivity is also reduced to a certain extent within this range. Due to the influence of the experimental instruments and the experimental environment, there is a lot of noise in the original spectral curve extracted from the ROI, which will affect the data quality and is not conducive to subsequent analysis and modeling. Therefore, in this paper, multiple scattering correction (MSC) and standard normal variable (SNV) methods are combined for preprocessing to eliminate the effects of high-frequency noise and baseline offset. Through pretreatment, the average spectral curves of healthy samples and infected samples of different types of roses can be seen more clearly in Figure 8b.

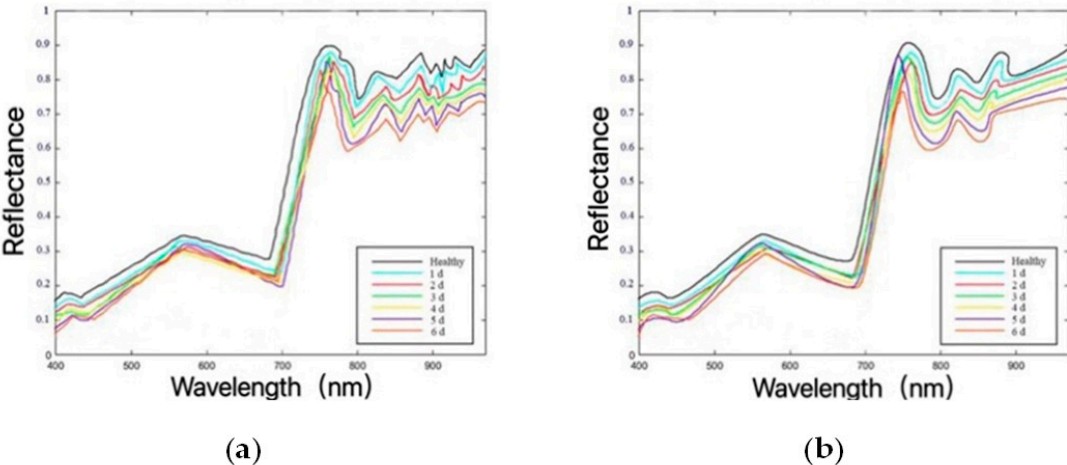

**Figure 8.** (**a**) Raw mean spectral of the healthy and infected samples and (**b**) mean spectral preprocessed by multiple scattering correction (MSC) + standard normal variable (SNV).

## 3. Results and Discussion

### 3.1. Optimizer Algorithm in the CNN Model

The optimizer algorithm is often used to find the optimal solution of the model. In this study, when using the CNN model to train the fully connected layer, the stochastic gradient descent (SGD) optimizer is used. It can use information more effectively when it is redundant, and the early iteration effect is excellent. A large number of theoretical and practical work proves that SGD can converge well in most instances. Especially when applying large data sets, training is performed at high speed. This study trains the SGD optimizer at different learning rates. Momentum is the historical gradient weight coefficient, set to 0.9, where the batch size is set to 32, and two iterations are performed first. The loss function is also the most critical element in model training. It also needs to be defined and optimized. The smaller the loss, the better the model. Taking the image feature in VGG16 as an example, we compare 5 loss functions. Table 2 shows the comparative results.

**Table 2.** The loss and accuracy results of different loss functions.

| Loss Function | Train_Loss | Train_Accuracy (%) | Test_Loss | Test_Accuracy (%) |
|---|---|---|---|---|
| categorical_crossentropy | 0.2231 | 95.63 | 0.2277 | 92.95 |
| mean_squared_error | 0.0910 | 92.05 | 0.1107 | 89.50 |
| mean_absolute_error | 0.1433 | 90.84 | 0.2122 | 85.75 |
| mean_squared_logarithmic_error | 0.0464 | 87.12 | 0.0557 | 83.95 |
| hinge | 0.6555 | 85.83 | 0.7040 | 80.10 |

All 5 loss functions reach an accuracy of over 80% and the train loss and test loss were controlled to a level lower than 0.23, which is a relatively good performance. Table 2 shows that when the loss

function is categorical_crossentropy, the train accuracy and test accuracy are both higher than the other loss functions. Therefore, this paper selects categorical_crossentropy as the loss function in the fully connected layer. Simultaneously, under the premise of two iterations, the appropriate learning rates were chosen as shown in Table 3.

**Table 3.** The loss and accuracy results of different learning rates.

| Learning Rate | Train_Loss | Train_Accuracy (%) | Test_Loss | Test_Accuracy (%) |
|---|---|---|---|---|
| 0.00009 | 0.2994 | 86.50 | 0.3269 | 83.75 |
| 0.0001 | 0.2523 | 90.50 | 0.2810 | 87.10 |
| 0.0002 | 0.2485 | 88.92 | 0.2903 | 87.30 |
| 0.0003 | 0.2766 | 87.87 | 0.2854 | 99.05 |
| 0.0004 | 0.2231 | 98.63 | 0.2277 | 97.75 |
| 0.0005 | 0.2599 | 92.68 | 0.2429 | 92.40 |
| 0.0006 | 0.2352 | 96.87 | 0.3297 | 90.95 |
| 0.001 | 0.2633 | 98.35 | 0.2528 | 93.30 |
| 0.01 | 0.6940 | 79.53 | 0.6932 | 80.00 |

It can be seen from Table 3 that with the increase of the learning rate, there is no obvious law for the changes of the other parameters. When the learning rate is set to 0.0004, better results can be achieved. The train accuracy and test accuracy are both highest while the train loss and test loss are lowest. In addition, we increased the number of iterations for training, and the results are shown in Figure 9.

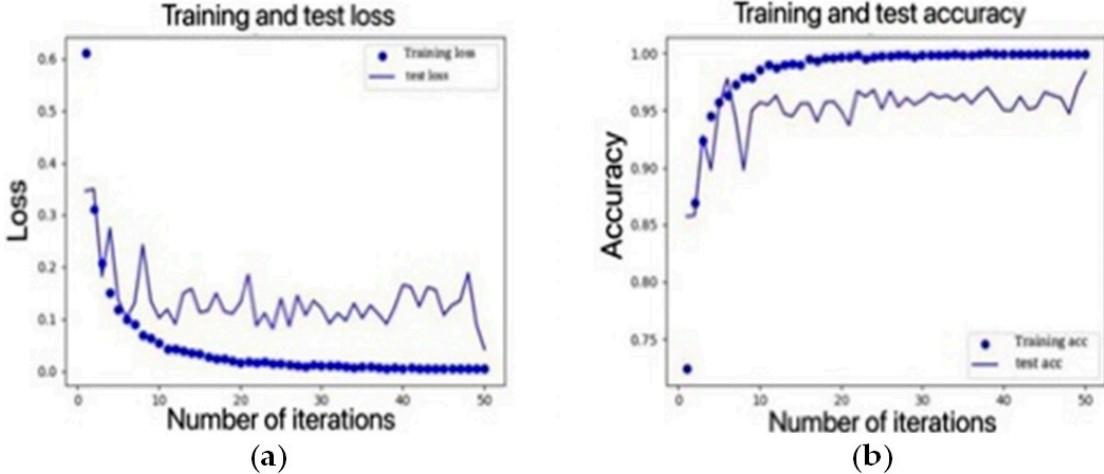

**Figure 9.** (**a**) The training and test loss of different iterations and (**b**) the training and test accuracy of different iterations.

It can be seen from Figure 9 that when the number of iterations is increased, the loss and accuracy of training increase or decrease regularly, the loss and accuracy of test show repeated fluctuations. The optimal accuracy of the training set can reach 99.60%, the loss is 0.0154, and the accuracy of the test set is 97.20%, with a loss of 0.0820. This shows that the adopted network can learn the features of the samples in more detail and accuracy.

*3.2. The Test Result of CNN*

Table 4 shows the classification accuracy of the three CNN models. It can be seen that the accuracy of detection for the two kinds of roses was above 80% for both in AlexNet. After MSC and SNV preprocessing, the accuracy of the training set and the test set have been improved to a certain extent, and the detection accuracy of 12–26 has reached a maximum of 100%, indicating that denoising preprocessing can effectively improve the accuracy of the model. At the same time, it can be seen from the table that under the spectral

feature, for 12–26 after SNV treatment, and for 13–54 after MSC treatment, the result is better, so in the subsequent modeling, different pretreatments were taken for the two roses. From VGG16 we can see the detection results based on image feature. After image preprocessing, the accuracy of the training set and the test set has also been improved, and the maximum value appears on the detection of 12–26, which is 99.6%. Based on the image feature, the detection accuracy of the susceptible variety is slightly higher than that of resistant variety as a whole, probably because the bacteria infect the susceptible variety faster and the changes reflected in the spectrum perform more obviously. Finally, according to the processing process of the two features, the two varieties are pretreated differently, and the NDDR detection model is established based on the fusion features. It can be seen that the detection accuracy based on the fusion features has reached more than 95%, which is higher than the previous two methods. Compared with AlexNet and VGG16, it can be seen that the fusion features perform better than the single feature. The model detection results are better for two reasons. One is because the hyperspectral data based on spectral and image features contain rich information from the samples, which reflects the characteristics of multiple features in the input to the network. Another reason is that the NDDR hierarchical structure is more abundant, which makes the model perform well in multi task feature fusion and feature dimension reduction, which can effectively extract features and prevent the disappearance of the lower gradient.

**Table 4.** The classification accuracy of the three models to detect black spot.

| AlexNet | | | |
|---|---|---|---|
| **Variety** | **Data Set** | **Train_Accuracy (%)** | **Test_Accuracy (%)** |
| 12–26 (Susceptible) | Raw | 92.36 | 89.58 |
| | Raw + MSC | 96.53 | 95.83 |
| | Raw + SNV | 100 | 97.92 |
| 13–54 (Resistant) | Raw | 87.50 | 83.33 |
| | Raw + MSC | 95.83 | 93.75 |
| | Raw + SNV | 94.44 | 91.67 |
| VGG16 | | | |
| Variety | Data Set | Train_Accuracy (%) | Test_Accuracy (%) |
| 12–26 (Susceptible) | Raw | 97.40 | 90.95 |
| | Raw + preprocessing | 99.60 | 97.20 |
| 13–54 (Resistant) | Raw | 97.10 | 93.53 |
| | Raw + preprocessing | 98.80 | 97.12 |
| NDDR-CNN | | | |
| Variety | Data Set | Train_Accuracy (%) | Test_Accuracy (%) |
| 12–26 (Susceptible) | Raw | 98.50 | 98.95 |
| | Raw + SNV + preprocessing | 100.00 | 99.63 |
| 13–54 (Resistant) | Raw | 97.87 | 96.57 |
| | Raw + MSC + preprocessing | 99.95 | 99.10 |

Therefore, the NDDR-CNN model based on the fusion feature has the best detection result for rose spot disease in this paper. The CNN model can well solve the high-dimensional and nonlinear practical problems of hyperspectral data. It effectively improves the detection results and avoids over-learning and under-learning. This research has achieved the early non-destructive detection of rose spot disease, and provides a basis for pathological detection of other plants. In the future, more work is needed on the research of plant pathological detection based on hyperspectral images.

## 4. Conclusions

The development of the forestry and flower industries requires effective identification of plant diseases, but the traditional machine learning recognition model based on a single feature has low accuracy, low efficiency and strong randomness. Based on these problems, this study explored the early non-destructive detection of rose spot disease based on the CNN model. In the establishment of the detection model, hyperspectral data and image preprocessing methods were introduced, and the

spectral and image features of the two rose leaves were extracted. For the small number of samples in the detection of black spot disease, CNN was applied to the study and three kinds of network structures were constructed. The effects of CNN's loss function, learning rate and different initialization methods on network performance were analyzed. Combined with the NDDR strategy, the accuracy of the detection model was improved, and the effectiveness of the preprocessing and feature extraction methods was verified. All the three models performed well; the results show that the NDDR-CNN model based on the fusion feature detected different types of roses 12–26 (100%) and 13–54 (99.95%), with the highest correlation coefficient with the real results. Further work will combine the physical and chemical indexes and microstructure of rose leaves to establish correlations with hyperspectral images, explain the changes in the spectrum from a biological point of view, and establish a more effective and accurate detection model.

**Author Contributions:** Conceptualization, J.M. and L.Y.; methodology, J.M. and L.P.; software, J.M.; validation, J.M. and L.P.; formal analysis, L.Y.; investigation, L.P. and L.Y.; resources, L.Y.; data curation, J.M., L.P. and L.Y.; writing—original draft preparation, J.M.; writing—review and editing, J.M., L.P. and L.Y.; visualization, J.M.; supervision, L.Y. and J.X.; project administration, L.P. and L.Y.; funding acquisition, L.Y. and J.X. All authors have read and agreed to the published version of the manuscript.

**Funding:** This research was funded by the National Natural Science Foundation of China [31770769] and the Fundamental Research Funds for the Central Universities [NO.2015ZCQ-GX-03].

**Conflicts of Interest:** The authors declare no conflict of interest.

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
