# Peer review of "Detection of Black Spot of Rose Based on Hyperspectral Imaging and Convolutional Neural Network"

_agriengineering, doi:10.3390/agriengineering2040037_

Round 1

Reviewer 1 Report

In this manuscript, authors try 3 different CNN models to detect Rose black spot through Hyperspectral imaging. The experiments were well designed. Nevertheless, the figure quality requires to be greatly improved such as but not limit to: 1) it’s hard to identify the details in Fig. 2; 2) in Fig. 6, what exactly do the spectral/image/fusion features stand for? Some typical examples will be helpful. 3) To explain the diagram of the 3 CNN models more intuitive, specific examples will be helpful. Moreover, there exists inconsistent description about the optimal CNN model. In Abstract, VGG16 was mentioned as the optimal model. However, in Conclusion Section, NDDR was described as the best one.

Author Response

Dear Reviewer:
Thank you for your letter and comments concerning our manuscript. Those comments are all valuable and very helpful for revising and improving our paper, as well as the important guiding significance to our researches. We have studied comments carefully and have made correction which we hope meet with approval. Revised portion are clearly highlighted in the manuscript. The response to your comments are in the attachment.

We tried our best to improve the manuscript and made some changes. We appreciate for your warm work earnestly, and hope that the correction will meet with approval.
Once again, thank you very much for your comments and suggestions.

Reviewer 2 Report

Dear Author/s,

I enjoyed reading your manuscript on the application of HSI for detection of black spot on rose leaves. Results seem interesting and may be applicable in future. However, authors are not sure about the science of the black spot in rose which I presume it is the primary knowledge or fact which should have been checked before submitting the work. The authors mention black spot is caused by "a kind of fungus pathogen" (line 25) and then contradict their statement by saying "bacteria infect the susceptible variety faster..." (line 214). Therefore, I presume the manuscript presents a mere classification of healthy and diseased leaves without proper scientific knowledge of the disease and how the technology can benefit the rose sector. Hence, the novelty present in the manuscript is a mere mathematical classification without a solid understanding of the background. Therefore, I recommend the editor to REJECT the manuscript. 

My general comments to the authors: 

Authors have used a lot of imprecise words such as "a kind of pathogen", "middle of each plant" and "suitable grow environment". These have terms could be translated through precise terms. 

The methodology section is poorly explained and is very difficult to replicate the study from the given description. Such as what a reader can understand by the middle of each plant - it definitely relates to height but what height, similarly, grew completely - what is the definition of completely. 

A suggestion, the use of hyperspectral imaging in Black spot could be in identifying latent stage and preventing the further spread. However, the study does not provide any insight. 

Thank you.

Author Response

(The authors gave the same response as above.)

Round 2

Reviewer 1 Report

All my concerns have been addressed in the revised manuscript. 

Author Response

Dear Reviewer,

We are very happy for receiving your reply, and quite honored to respond your concerns in the revised manuscript. Your suggestions are very valuable to us.

Thank you again for taking the time to review our manuscript. 

Best regards to you!